# Learning Interatomic Potentials at Multiple Scales

**Xiang Fu**
MIT CSAIL
`xiangfu@mit.edu`

**Albert Musaelian**
Harvard John A. Paulson
School of Engineering and Applied Sciences
`albym@g.harvard.edu`

**Anders Johansson**
Harvard John A. Paulson
School of Engineering and Applied Sciences
`andersjohansson@g.harvard.edu`

**Tommi Jaakkola**
MIT CSAIL
`tommi@csail.mit.edu`

**Boris Kozinsky**
Harvard John A. Paulson School of Engineering and Applied Sciences and
Robert Bosch Research and Technology Center
`bkoz@g.harvard.edu`

## Abstract

The need to use a short time step is a key limit on the speed of molecular dynamics (MD) simulations. Simulations governed by classical potentials are often accelerated by using a multiple-time-step (MTS) integrator that evaluates certain potential energy terms that vary more slowly than others less frequently. This approach is enabled by the simple but limiting analytic forms of classical potentials. Machine learning interatomic potentials (MLIPs), in particular recent equivariant neural networks, are much more broadly applicable than classical potentials and can faithfully reproduce the expensive but accurate reference electronic structure calculations used to train them. They still, however, require the use of a single short time step, as they lack the inherent term-by-term scale separation of classical potentials. This work introduces a method to learn a scale separation in complex interatomic interactions by co-training two MLIPs. Initially, a small and efficient model is trained to reproduce short-time-scale interactions. Subsequently, a large and expressive model is trained jointly to capture the remaining interactions not captured by the small model. When running MD, the MTS integrator then evaluates the smaller model for every time step and the larger model less frequently, accelerating simulation. Compared to a conventionally trained MLIP, our approach can achieve a significant speedup (~3x in our experiments) without a loss of accuracy on the potential energy or simulation-derived quantities.

## 1 Introduction

The interatomic potential energy that governs the dynamics of a system of atoms has long been both understood and modeled as a combination of atomic interactions of various strengths and scales. In a system containing a comparatively stiff molecule in a soft fluid, for example, the intramolecular forces are much stronger than the intermolecular forces from the solvent. Classical potentials, such as the popular optimized potentials for liquid simulations (OPLS, [1, 2]), explicitly define the potential as such a sum over simple analytic terms:

$$E(\boldsymbol{r}) = E_{\text{bonds}}(\boldsymbol{r}) + E_{\text{angles}}(\boldsymbol{r}) + E_{\text{dihedrals}}(\boldsymbol{r}) + E_{\text{nb}}(\boldsymbol{r})$$

where $r$ denotes the atom positions; $E_{\text{bonds}}(r)$, $E_{\text{angles}}(r)$, and $E_{\text{dihedrals}}(r)$ are intramolecular (or "bonded") bond, angle, and torsional potentials; and $E_{\text{nb}}$ denotes the nonbonded, including intermolecular potential term. Rigid interactions such as bond vibrations, governed by $E_{\text{bonds}}$, occur at fast time scales, while the nonbonded interactions $E_{\text{nb}}$ are slower and smoother. Due to the greater number of atoms involved in nonbonded interactions, however, the computational expense of calculating the $E_{\text{nb}}$ term can be meaningfully larger.

Essentially, the intramolecular and intermolecular nonbonded forces are of different scales. While the intramolecular forces require a short time step, integrating the intermolecular forces at the same step size is often overkill. To harness this separation of both scales and computational cost, multiple-time-step (MTS) integrators [3–7] integrate the fast-evolving terms with a short time step and the slow-evolving terms with a long time step. MTS integrators are theoretically principled and have been widely used in various classical MD workflows [8–12], usually bringing a speedup of two to four times.

The approximations and limited functional forms of classical potentials are not sufficient for many applications. Ab initio molecular dynamics (AIMD) simulations—governed by a potential energy surface computed with electronic structure methods such as Density Functional Theory (DFT)—are widely used but suffer from dramatic computational limitations on time- and length-scale due to the expense and unfavorable scaling of the electronic structure calculations. The application of MTS schemes to AIMD has been limited by the difficulties in decomposing the ab initio potential into components of separate scales in the absence of a simple analytical form [13, 14].

Machine learning interatomic potentials (MLIPs) [15–48] are increasingly used to run MD simulations orders of magnitude cheaper than AIMD while preserving near-AIMD accuracy. MLIPs also largely avoid classical potentials' assumption of an explicit discrete covalent bond topology, which greatly broadens their applicability, notably in materials science and reactive simulations. While MLIPs have enabled many previously impractical or impossible simulations, further improvements in the speed and cost of MLIP-driven MD are extremely valuable. MTS methods remain largely unexplored for MLIPs: although recent research [49] has combined an infrequently evaluated MLIP with a classical potential evaluated at every time step, a methodology for machine learning a scale-separated potential model from data is, to the best of our knowledge, lacking in the literature. This work presents an approach for learning complementary scale-separated MLIPs from data, yielding a ~3x speedup in MD simulations using MTS integration.

## 2   Preliminaries

**MD simulation** involves integrating a Newtonian equation of motion $\ddot{r} = d^2 r/dt^2 = m^{-1} F(r)$ with atomic positions $r$, masses $m$, and forces $F$. The forces are obtained by differentiating the potential energy of a molecular system $E(r)$ with respect to the atomic positions $r$: $F(r) = -\partial E/\partial r$.

**Allegro** [37] is an $E(3)$-equivariant [50] neural network MLIP architecture for predicting $E(r)$ that enforces strictly local interactions while achieving state-of-the-art performance. Its strict locality enables efficient parallel scaling across GPUs to reach larger length- and time-scales [51]. In Allegro, every pair of atoms within a chosen fixed cutoff distance $r$ are considered neighbors. For each ordered pair of neighboring atoms $i$ and $j$, an Allegro model produces a learnable, $E(3)$-invariant, many-body final latent representation $x^{ij}$ of the geometry of $i$, $j$, and all other neighbors of $i$. An edge energy $E_{ij}$ is then predicted from it via the output block, a multi-layer perception $\text{MLP}_{\text{out}}$ without bias; the sum over edge energies gives the total potential energy:

$$E_{\text{ML}}(r) = \sum_{(i,j):||r_i - r_j|| \leq r} E_{ij} = \sum_{(i,j):||r_i - r_j|| \leq r} \text{MLP}_{\text{out}}(x^{ij}) \qquad (1)$$

An Allegro model's predicted $E_{\text{ML}}(r)$ is trained to reproduce the potential energy and forces from a reference method such as DFT using a loss function like:

$$\mathcal{L} = \lambda_E \left\| \frac{E_{\text{ML}} - E}{N} \right\|^2 + \lambda_F \frac{1}{3N} \left\| -\frac{\partial E_{\text{ML}}}{\partial r} - F \right\|^2 \qquad (2)$$

where $\lambda_E$ and $\lambda_F$ are loss coefficients for energy and forces and $N$ is the number of atoms. For complete details on Allegro and its training see [37].

**MTS Integrators** accelerate MD simulations by propagating different parts of the dynamics with different time steps suited to their respective characteristic time scales. In this paper, we focus on the reversible reference system propagator algorithms (rRESPA [5, 52]), an MTS integrator popular for its rigorous derivation, time-reversibility, and symplectic properties. We show the rRESPA algorithm in Algorithm 1 and refer interested readers to [5] for detailed derivations.

---

**Algorithm 1** An integration step of the MTS integrator (rRESPA)

---

1: **Input:** Inner time step $\Delta t$, number of inner time steps per outer time step $N_{\text{inner}}$, short-range force $\boldsymbol{F}_s$, long-range force $\boldsymbol{F}_l$, atom masses $\boldsymbol{m}$, initial positions $\boldsymbol{r}$, initial velocities $\dot{\boldsymbol{r}}$
2: $\dot{\boldsymbol{r}} \leftarrow \dot{\boldsymbol{r}} + \frac{1}{2}(N_{\text{inner}}\Delta t) \cdot \boldsymbol{m}^{-1}\boldsymbol{F}_l(\boldsymbol{r})$
3: **for** step $i = 1, \ldots, N_{\text{inner}}$ **do**
4: $\quad$ $\dot{\boldsymbol{r}} \leftarrow \dot{\boldsymbol{r}} + \frac{1}{2}\Delta t \cdot \boldsymbol{m}^{-1}\boldsymbol{F}_s(\boldsymbol{r})$
5: $\quad$ $\boldsymbol{r} \leftarrow \boldsymbol{r} + \dot{\boldsymbol{r}}\Delta t$
6: $\quad$ $\dot{\boldsymbol{r}} \leftarrow \dot{\boldsymbol{r}} + \frac{1}{2}\Delta t \cdot \boldsymbol{m}^{-1}\boldsymbol{F}_s(\boldsymbol{r})$
7: **end for**
8: $\dot{\boldsymbol{r}} \leftarrow \dot{\boldsymbol{r}} + \frac{1}{2}(N_{\text{inner}}\Delta t) \cdot \boldsymbol{m}^{-1}\boldsymbol{F}_l(\boldsymbol{r})$

---

## 3 Learning Scale Separation

To achieve scale separation and harness the MTS integrator, we need to enable efficient calculation of stiff and fast-evolving force terms. Meanwhile, we still need to capture the smooth and slow-evolving force terms so that the overall machine learning potential remains accurate. For many molecular systems of interest, short-range interactions, such as covalent bonds, are strong and induce stiff motions, while long-range interactions, such as non-bonded interactions, can be integrated with a longer time step.

The above observation motivates separating scales by combining MLIPs with different receptive fields. Allegro's unique combination of the leading accuracy of equivariant techniques with strict locality particularly lends itself to such a scale separation scheme. We train two models:

- **Inner model** ($E_{\text{inner}}$): An efficient model with fewer parameters and interaction layers and a small radial cutoff that captures short-time-scale interactions.

- **Outer model** ($E_{\text{outer}}$): An expressive model with a larger number of parameters and interaction layers and a larger radial cutoff that fits the remaining interactions not learned by the inner model.

We let the two models jointly predict the potential energy: $E_{\text{ML}}(\boldsymbol{r}) = E_{\text{inner}}(\boldsymbol{r}) + E_{\text{outer}}(\boldsymbol{r})$. Training the two models together from scratch, however, may not induce scale separation: the outer model has sufficient capacity to learn the entire reference potential energy surface, and so the short-range interactions may not be attributed to the inner model. To avoid such degeneracy, at the beginning of training, we first freeze all parameters in the outer model to let the inner model fit the force and energy within its capacity. We then later start training the outer model with a zero-initialization over its output block $\text{MLP}_{\text{out}}$. This zero initialization ensures $E_{ij}^{\text{outer}} = 0$ and $\partial E_{ij}^{\text{outer}}/\partial\boldsymbol{r} = 0$ for all atom pairs $(i, j)$, preventing the initial noise of the outer model from interfering with the interactions already learned by the inner model. Zero-initialization has been widely used in previous works for finetuning pretrained models [53]. The training procedure is presented in Algorithm 2. We refer to our scale-separated Allegro model as **MTS-Allegro**.

## 4 Experiments

Our experiments consider an ab initio water system [54]. This dataset contains 1593 reference calculations of bulk liquid water at the revPBE0-D3 level of accuracy. Each structure contains 192 atoms (64 water molecules). We randomly sample 1000 structures for training, 100 structures for validation, and the rest for testing[1]. Both the Allegro model and the outer model of MTS-Allegro have

---

[1]These structures are not sampled under an equilibrium condition such as NVE or NVT ensembles.

**Algorithm 2** Learning MLIPs at multiple scales
_________________________________________________________________________________
1: **Input**: Inner model $E_{\text{inner}}$ and outer model $E_{\text{outer}}$, inner training number of epoch $M_{\text{inner}}$, total training number of epoch $M$
2: **Output**: Trained models $E_{\text{inner}}$ and $E_{\text{outer}}$
3: Zero-initialize the outer model's output layer: $\text{MLP}_{\text{out}}$ of $E_{\text{outer}}$
4: **for** epoch $i = 1, \ldots, M_{\text{inner}}$ **do**
5:     train $E_{\text{ML}} = E_{\text{inner}}$ through the loss defined in Equation (2)
6: **end for**
7: **for** epoch $i = M_{\text{inner}} + 1, \ldots, M$ **do**
8:     train $E_{\text{ML}} = E_{\text{inner}} + E_{\text{outer}}$ through the loss defined in Equation (2)
9: **end for**
_________________________________________________________________________________

Table 1: Energy and Force prediction mean absolute error (MAE) and root mean square error (RMSE).

|  | Allegro | MTS-Allegro | MTS-Allegro, inner |
|---|---|---|---|
| Energy MAE [meV/Atom] | 2.4 | 1.6 | 12.5 |
| Forces MAE [meV/Å] | 40.3 | 35.5 | 72.5 |
| Forces RMSE [meV/Å] | 77.4 | 76.9 | 118.3 |

two interaction layers, a 6 Å cutoff, and the same parameter count; the inner model of MTS-Allegro has one interaction layer, a 4 Å cutoff, and half the width in each layer compared to the outer model. Detailed hyperparameters are included in Appendix A.

We compare the accuracy in recovering the reference potential and various simulation-derived quantities between (1) our proposed MTS-Allegro model with rRESPA integration, (2) a conventionally trained Allegro model with standard Velocity Verlet integration, and (3) the inner model of MTS-Allegro alone with Velocity Verlet integration. We use a time step of $0.5$ fs for Velocity Verlet integration and the inner loop of MTS integration, which is standard for water simulations [55–57]. For MTS-Allegro, we experiment with outer time steps of $[1.0, 2.0, 3.0, 4.0]$ fs, corresponding to [2x, 4x, 6x, 8x] multiples of the inner time step.

**Force and energy prediction accuracy.** We report the force and energy prediction errors in Table 1. The MTS-Allegro model has a very similar performance to a conventionally trained Allegro model. Unsurprisingly, the inner model of MTS-Allegro alone obtains higher errors due to its limited receptive field and capacity.

**Structure and dynamics.** To investigate MTS-Allegro's ability to reproduce structural and dynamical observables, we simulate the water system in a canonical (NVT) ensemble (constant volume and temperature) at 300 K and compute the element-wise radial distribution functions (RDFs, structural) and the mean squared displacement (MSD, dynamical). We simulate for 400 picoseconds (ps) and remove the first 50 ps for equilibration when computing the observables. All models remain stable throughout the entire simulation. Figure 1 (a-d) shows the RDFs and MSD (in log-log space) for Allegro, the inner model of MTS-Allegro, and MTS-Allegro under different outer step sizes. MTS-Allegro simulations with 2x to 8x outer time steps all achieve excellent agreement with Allegro on the RDFs and MSD, while the inner model alone produces erroneous dynamics and observables due to its limited capacity and, thereby, accuracy in recovering the potential.

**Energy conservation under the microcanonical ensemble.** For a correct MD simulation in the microcanonical (NVE) ensemble, the total energy (sum of potential and kinetic energies) should remain constant over time. To evaluate the energy conservation property of the MLIPs, we initialize the water system using a structure from the test dataset and a temperature of 300 K, run energy minimization, and then simulate for 100 ps in the NVE ensemble. Figure 1 (e) shows the drift of total energy from the first frame (after removing the first 20 ps for equilibration). We observe that Allegro and MTS-Allegro with 2x and 4x outer time steps are energy-conserving. MTS-Allegro with 6x and 8x outer time steps yield drifts of $0.12$ meV/(atom·ps) and $0.18$ meV/(atom·ps), respectively.

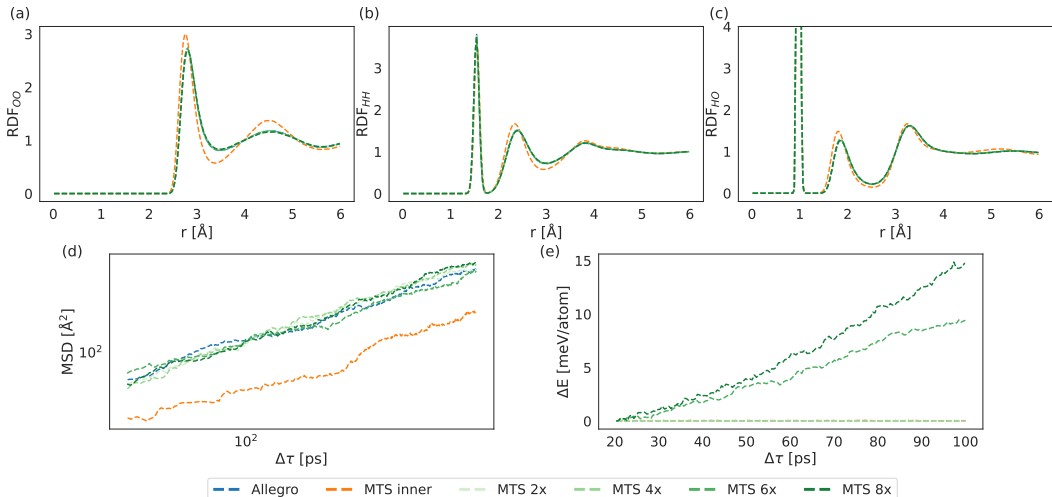

Figure 1: (a-c) O-O, H-H, and H-O RDFs of NVT simulations. (d) MSD of NVT simulations. (e) Total energy drift of NVE simulations. In the legend, MTS-Allegro is shortened for "MTS" along with the outer-inner time step ratio.

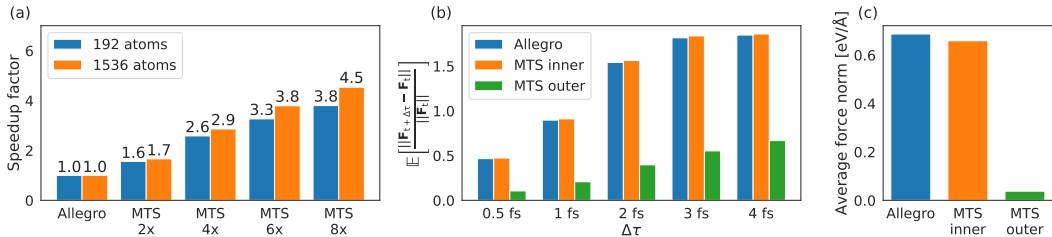

Figure 2: (a) Relative speedup factor compared to Allegro (1x in the plot). (b) Average relative frame-to-frame force difference in $0.5$ fs time step Velocity Verlet MD for different models. (c) Average norm of forces for different models.

**Speedup.** Figure 2 (a) shows the relative speedup of MTS-Allegro compared to Allegro at two system sizes as measured in MD simulation[2]. MTS-Allegro with a 4x outer time step, which preserves simulation-derived quantities and maintains energy conservation, achieves an MD speedup of 2.6x for a system size of 192 atoms and 2.9x for a system size of 1536 atoms. MTS-Allegro 6x and 8x, which obtain a further speedup, remain reliably stable in our experiments and faithfully reproduce RDFs and MSDs but do not maintain energy conservation. Such larger outer time step simulations may still find use in preliminary or other calculations whose requirements are less stringent.

**Analysis of scale separation.** We investigate how well MTS-Allegro separates scales by inspecting the time-smoothness and strength of the learned interactions. Figure 2 (b) shows the average relative frame-to-frame force difference (defined as $\mathbb{E}_t\left[\|\boldsymbol{F}_{t+\Delta\tau} - \boldsymbol{F}_t\| / \|\boldsymbol{F}_t\|\right]$ for each component) in $0.5$ fs time step Velocity Verlet MD for different models. Compared to the conventional Allegro model and the inner model, the outer model of MTS-Allegro outputs forces with a much lower $\mathbb{E}_t\left[\|\boldsymbol{F}_{t+\Delta\tau} - \boldsymbol{F}_t\| / \|\boldsymbol{F}_t\|\right]$ for all $\Delta\tau$, indicating that it learns interactions that change much more slowly in time. Figure 2 (c) shows the average force norm (defined as $\mathbb{E}_t\left[\|\boldsymbol{F}_t\|\right]$) of different models. The outer model learns interactions that are much weaker in magnitude than Allegro and the inner model of MTS-Allegro. Weak and slow-varying interactions are exactly what allows for a longer time step. This analysis confirms the effectiveness of MTS-Allegro in learning scale separation.

---

[2]All simulations were run in LAMMPS [12] using `pair_allegro` on a single NVIDIA Tesla V100-PCIe 32GB GPU. The speed of Allegro is $0.51$ and $0.072$ ns/day for 192 and 1536 atoms, respectively.

## 5 Conclusion

We have developed a method to accelerate MLIP-driven MD simulations by learning a scale separation and using an MTS integrator. In an ab initio water system, our approach achieves around three times speedup without loss in accuracy on the potential energy or simulation-derived quantities. MTS integrators can also be used with more than two levels, and learning finer-grained scale separations with more than two MLIPs is a direction for future work. It is also possible to co-train different model architectures, such as kernel-based methods [18] and message-passing MLIPs [36], for accuracy-speed trade-offs. The presented technique promises a direction for significant practical speed gains when running MLIP-driven MD, including in large and complex systems.

## Acknowledgments and Disclosure of Funding

We thank Simon Batzner, Cameron Owen, Yilun Xu, and Wujie Wang for valuable feedback and insightful discussions. X. F. and T. J. acknowledge support from the MIT-GIST collaboration. The work at Harvard was supported by Bosch Research, DOE Office of Basic Energy Sciences Award No. DE-SC0022199 and the Harvard University Materials Research Science and Engineering Center Grant No. DMR-2011754. A. M. was supported by DOE, Scientific Computing Research, Computational Science Graduate Fellowship under Award Number DE-SC0021110. Computing resources were provided by the MIT SuperCloud and Lincoln Laboratory Supercomputing Center and the Harvard University FAS Division of Science Research Computing Group.

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

# A  Supplementary Information

Table 2: Hyperparameters for Allegro and the outer model of MTS-Allegro.

| Hyperparameter | Value |
|---:|:---|
| two body MLP latent dimensions | $[128, 256, 512, 1024]$ |
| latent MLP latent dimensions | $[1024, 1024, 1024]$ |
| number of interaction layers | 2 |
| radius cutoff | 6 Å |
| maximum rotation order ($l_{\max}$) | 2 |
| atom embedding multiplicity | 32 |

Table 3: Hyperparameters for the inner model of MTS-Allegro.

| Hyperparameter | Value |
|---:|:---|
| two body MLP latent dimensions | $[64, 128, 256, 512]$ |
| latent MLP latent dimensions | $[512, 512, 512]$ |
| number of interaction layers | 1 |
| radius cutoff | 4 Å |
| maximum rotation order ($l_{\max}$) | 2 |
| atom embedding multiplicity | 32 |

Table 4: Hyperparameters for training. Both Allegro and Allegro-MTS use the same set of training hyperparameters (while $M_{\mathrm{inner}}$ is only applicable to Allegro-MTS).

| Hyperparameter | Value |
|---:|:---|
| batch size | 1 |
| optimizer | Adam |
| initial learning rate | 0.005 |
| learning rate scheduler | ReduceLROnPlateau |
| learning rate patience | 5 epochs |
| learning rate factor | 0.5 |
| early stopping learning rate | $10^{-6}$ |
| $\lambda_E$ | 1 |
| $\lambda_F$ | 2 |
| inner only epochs ($M_{\mathrm{inner}}$ in Algorithm 2) | 20 epochs |

