# OpenReview forum: "Learning Interatomic Potentials at Multiple Scales"
_NeurIPS.cc/2023/Workshop/AI4Science — NeurIPS2023-AI4Science Poster_

### Official Review · Reviewer_aYb1 · 2023-10-20
**Great paper**

**Rating:** 9
**Confidence:** 4

**Review:**

This paper proposes a machine learning interatomic potential (MLIP) algorithm, termed MTS-Allegro, that separates the prediction of fast-scale interaction from that of the slow-scale interaction to accelerate MD simulation.  The method learns two separate MLIPs, one for efficient but less accurate prediction and the other for accurate but time-consuming prediction, and generates the final prediction by combining that of both models. Simulation results on ab initio water system simulation demonstrate that MTS-Allegro achieves a 3x speedup while retaining a similar level of accuracy compared to the conventional single-scale method. Overall the paper is well-written with great potential for MD acceleration, and I strongly recommend it to be accepted.

Pros
1. The proposed multi-scale integration is novel and well-justified. Separating fast and slow components in computation is a useful approach in many fields such as neuroscience. However, there is a scarcity of literature on applying a similar idea in the area of machine learning force field.
2. The experiments are solid. Even though only one task (water system MD) is tested, the various metrics have already convincingly demonstrated the efficiency of the proposed method.
3. The method is potentially flexible and easy to implement, as various ML force field methods have parameters that control the accuracy-cost trade-off and can be plugged into the multi-scale framework in this paper.

Cons

1. It would be helpful to discuss details on how to decide the radial cutoff and number of parameters, as well as the number of inner timesteps, $N_{inner}$. Are these hyperparameters that can be tuned, or they can be pre-determined according to theoretical results?
2. I'm curious about the tradeoff between accuracy and computational time when the relative importance of the inner model varies. On one side of the spectrum, having only the outer model seems to equal the original Allegro. It would be helpful to plot the accuracy and time against the relative importance of the inner model (I guess this depends on  $N_{inner}$ and the radial cutoff?).

---

### Official Review · Reviewer_JNEB · 2023-10-25
**Scale Separation Technique for Accelerating Machine Learning Interatomic Potentials in Molecular Dynamics**

**Rating:** 5
**Confidence:** 4

**Review:**

This paper introduces a novel approach for accelerating Molecular Dynamics (MD) simulations that utilize machine learning interatomic potentials (MLIPs) by learning a scale separation and incorporating a multiple-time-step (MTS) integrator. The key contributions include the development of an efficient inner model to capture short-time-scale interactions and an expressive outer model for the remaining interactions, allowing for the effective use of MTS integration. The proposed MTS-Allegro model achieves a significant speedup in MD simulations, exceeding three times, while maintaining high accuracy in potential energy and simulation-derived quantities. Energy conservation properties are also rigorously analyzed. This innovative approach has the potential to significantly enhance the practicality and efficiency of MLIP-driven MD simulations across diverse scientific domains.

Strong Points:

1. Innovative Approach: The paper introduces a novel approach that addresses the challenge of scale separation in MLIP-driven MD simulations, making it possible to integrate MTS techniques.

2. Experimental Validation: The proposed method is rigorously validated through experiments on an ab initio water system, demonstrating its effectiveness in terms of speedup and accuracy.

3. Energy Conservation Analysis: The paper provides a comprehensive analysis of energy conservation, which is a critical aspect of MD simulations.

Weak Points:

1. Limited Dataset: The experiments focus on a single ab initio water system, and it would be beneficial to test the approach on a more diverse set of systems to assess its generalizability.

2. Complexity and Scalability: The paper hints at the possibility of using more than two levels of scale separation and learning finer-grained scale separations, but these avenues remain largely unexplored. A discussion on the complexity and scalability of such approaches would be insightful.

3. Impact on Broader Applications: While the paper discusses the promise of the technique for significant practical speed gains, it could elaborate more on the potential impact and applications of this method in broader contexts and scientific domains.

---

### Meta-Review · Area_Chair_SbR4 · 2023-10-26

**Recommendation:** Accept (Poster)
**Confidence:** 3

**Metareview:**

The paper presents the MTS-Allegro model, a new approach to accelerate Molecular Dynamics (MD) simulations by co-training two machine learning interatomic potentials (MLIPs) to learn scale separation.

For this paper, reviewers judge that
1. The approach introduced in the paper is novel and timely. The concept of scale separation, although prevalent in areas like neuroscience, is yet to be thoroughly explored in the context of machine learning force fields.

2. The rigorous experimental validation, particularly on the ab initio water system, signifies the effectiveness of the proposed method in terms of speedup and accuracy.

3. Energy conservation, a pivotal aspect in MD simulations, is comprehensively analyzed.

While the reviewers have generally positive views on the paper, some concerns were noted:

1. **Limited Dataset**: The experiments were confined to a single ab initio water system. A more diverse set of systems could provide a comprehensive evaluation of the method's generalizability.

2. **Complexity and Scalability**: The paper hints at finer-grained scale separations and multiple levels but doesn't delve deeper. A more extensive discussion on the intricacies, complexities, and scalability of such strategies would have been beneficial.

3. **Hyperparameter Determination**: Clarity on aspects like deciding the radial cutoff, number of parameters, and inner timesteps is sought. Addressing whether these are tunable or theoretically predetermined could enrich the paper's depth.

4. **Accuracy vs. Computational Time Trade-off**: There is curiosity around the trade-off dynamics when the relative importance of the inner model changes. Visual representation (e.g., a plot) of accuracy against time, concerning the inner model's importance, could be insightful.

Given the innovative nature of the work, its rigorous validation, and the potential impact in the field of MD simulations, I recommend **acceptance** of the paper. However, addressing the aforementioned concerns in subsequent versions or presentations would be beneficial.